# Assessing antioxidant, antidiabetic potential and GCMS profiling of ethanolic root bark extract of *Zanthoxylum rhetsa* (Roxb.) DC: Supported by *in vitro, in vivo* and *in silico* molecular modeling

Apurba Kumar Barman[1]*, Sumaiya Mahadi[1], Md Arju Hossain[2], Rahima Begum[3], Rabindra Nath Acharyya[4], Marjana Alam[1], Md. Habibur Rahman [ID][5,6], Nripendra Nath Biswas[4], A. S. M. Monjur Al Hossain [ID][7]*

1 Department of Pharmacy, R. P. Shaha University, Naryanganj, Bangladesh, 2 Department of Microbiology, Primeasia University, Banani, Bangladesh, 3 Department of Pharmacy, Noakhali Science and Technology University, Noakhali, Bangladesh, 4 Pharmacy Discipline, Life Science School, Khulna University, Khulna, Bangladesh, 5 Department of Computer Science and Engineering, Islamic University, Kushtia, Bangladesh, 6 Center for Advanced Bioinformatics and Artificial Intelligent Research, Islamic University, Kushtia, Bangladesh, 7 Faculty of Pharmacy, Department of Pharmaceutical Technology, University of Dhaka, Dhaka, Bangladesh

* monjur.shiplu@du.ac.bd (ASMMAH); apurba_phr@rpsu.edu.bd (AKB)

## Abstract

*Zanthoxylum rhetsa (ZR)* is used traditionally to manage a variety of ailments, including diabetes. Oxidative stress may accelerate the diabetic condition. The available antidiabetic and antioxidant drugs have many shortcomings including resistance, inefficiency, higher dose, side effects and costs. The goal of the current investigation was to assess the antioxidant capacity and antidiabetic activity of an ethanolic extract of *Zanthoxylum rhetsa* root bark (ZRRB) through *in vitro*, *in vivo*, and *in silico* methods. The antioxidant capacity of the ZRRB extract was measured using both the DPPH radical assay and the total antioxidant activity test. The oral glucose tolerance test (OGTT) and alloxan-induced diabetic mice model were also used to examine *in vivo* antidiabetic efficacy. Phytochemicals identification was done by GCMS analysis. Additionally, computational methods such as molecular docking, ADMET analysis, and molecular dynamics (MD) modeling were performed to determine the above pharmacological effects. The extract demonstrated significant DPPH scavenging activity ($IC_{50} = 42.65$ µg/mL). In the OGTT test and alloxan-induced diabetes mice model, the extract effectively lowered blood glucose levels. Furthermore, *in vitro* inhibition of pancreatic α-amylase studies demonstrated the ZRRB extract as a good antidiabetic crude drug ($IC_{50} = 81.45$ µg/mL). GCMS investigation confirmed that the crude extract contains 16 major phytoconstituents, which were docked with human peroxiredoxin-5, α-amylase, and sulfonylurea receptor 1. Docking and pharmacokinetic studies demonstrated that among 16 phytoconstituents, 6H-indolo[3,2,1-de] [1,5]naphthyridin-6-one (CID: **97176**) showed the highest binding affinity to targeted enzymes, and imitated Lipinski's rule of five. Furthermore, MD simulation data confirmed that the aforementioned compound is very steady to the

**Data Availability Statement:** All the relevant data are within the paper and its Supporting Information files.

**Funding:** The author(s) received no specific funding for this work.

**Competing interests:** The authors have declared that no competing interests exist.

binding site of α-amylase and sulfonylurea receptor 1 receptors. Findings from *in vitro*, *in vivo and in silico* investigation suggest that ZRRB extract contains a lead compound that could be a potent source of antidiabetic drug candidate.

## Introduction

Since ancient times, man has been familiar with plants and has employed them in numerous ways throughout history. Many plants utilized as medicines were discovered initially due to the reciprocal relationship between humans and plants [1]. The World Health Organization (WHO) lists 255 medications as crucial and 11% of them are sourced from plants and also several synthetic drugs are derived from natural precursors [2].

Hyperglycemia (high blood sugar) and inadequate insulin synthesis characterize a chronic metabolic condition known as diabetes mellitus (DM). In the presence of chronic hyperglycemia, non-enzymatic glycosylation and diabetes complications may further be propagated [3]. The diabetic condition increases the risk of micro-vascular complications and atherosclerosis. The continuous exposure to reactive oxygen species (ROS) due to the absence of adequate antioxidants in the body may further worsen diabetes complications [4]. The present worldwide occurrence of diabetes is projected at 143 million, but it is anticipated to increase to 300 million by 2025 [5]. Managing the condition of diabetes with no side effects still remains a challenge, and the presently marketed antidiabetic drugs show a number of limitations including hypoglycemia (Sulphonylureas), lactic acidosis, vitamin B12 and folate malabsorption, gastrointestinal symptoms (Acarbose), overweight (Thiazolidinediones and Sulphonylureas), and even edema (Thiazolidinediones) [6]. Certain antioxidants may also block α-amylase activity, potentially to control postprandial hyperglycemia. The α-amylase and α-glucosidase inhibitory drugs slow carbohydrate metabolism and thus control blood sugar concentrations and also the underlying physiological disorder [7]. The sulfonylurea 1 receptor is an important protein that regulates insulin excretion in pancreatic $\beta$-cells indications to lower the blood sugar level [8]. On the other hand, the thiol-dependent peroxiredoxin antioxidant family member protein appears to be involved in signaling cascades that shield diabetic patients from oxidative damage and in scavenging-oxidants [9]. As a result, previously, many investigators conducted research on the possibility of bioactive compounds especially plants to modify the activity of these reported proteins.

*Zanthoxylum rhetsa* (Roxb.) DC., also known as Bajna or Cape yellowwood, is a medium-sized (10–13 m) tree in the Rutaceae family that is available in Bangladesh, Indonesia, India, China, Malaysia, as well as other tropical countries [10]. Numerous portions of the *Z. rhetsa* are traditionally employed in the management of numerous disorders such as anti-inflammatory, diuretic, anti-diabetic, and antispasmodic [11]. It also bears antidiarrheal and anti-nociceptive properties. Fruits and stem barks of the plant are used in the management of rheumatism, urinary tract infection, asthma, bronchitis, heart troubles, toothache, and also as an astringent, and stimulant. The essential oil has disinfecting, antiseptic, and anti-inflammatory properties, and it is effective against cholera [12]. Bark extract has been claimed to provide relief from stomach and chest aches, and to assist in treating snake bites [13]. Leaf extract obtained by decocting has been found effective against intestinal parasitic manifestations and insect attacks [13]. Paste derived from the hard spines of the plant is employed to ease the pain as well as boost up milk production in lactating mothers [14]. *Z. rhetsa* is used as a spice and condiment in Thai cuisine and as an infection remedy [15]. Different earlier phytochemicals

evaluations of this plant extract have displayed the existence of various compounds containing alkaloids, monolignols, coumarins, and lignans like 8-methoxy-N-methylflindersine, 3,5-dimethoxy-4-geranyloxycinnamyl alcohol, xanthyletin, and sesamin [16]. It has been reported that four quinolone terpene alkaloids such as chelerybulgarine, 20-episimulanoquinoline, 2, 11-didemethoxy-vepridimerine B, and rhetsidimerine were isolated from *Zanthoxylum rhetsa* [17] root bark. The plant has been shown to have various pharmacological activities like antimicrobial [18], anti-inflammatory [19], antioxidant [20], antidiarrheal [21], and anticancer activity against B16-F10 melanoma cancer cell line [22]. Methanolic and aqueous extracts of the stem bark of the plant have been reported to alleviate oxidative stress, dyslipidemia, hyperglycemia as well as diabetes in streptozocin-induced diabetic rats [23]. *Z. rhesta* leaf extracts in aqueous solvent possessed maximum anti-diabetic effect compared to 95% ethanol and methanol solvents. The extract also showed moderate anti-oxidant activity in these three solvents [11]. Essential oils extracted from ripe fruits of *Z. rhesta* have shown promising anti-diabetic, anti-gout activity as well as cytotoxic activity against Meg-01 leukemia cell line [24]. *Z. rhesta* root bark with two isolated compounds from four different fractions has been reported as having significant *in vitro* anti-oxidant, cytotoxic, anti-microbial, and thrombolytic activity [25]. *Z. rhesta* (Roxb.) nano-emulsion formulated from dried fruit extracts has been claimed to have promising anti-oxidant and anti-inflammatory properties for topical application [26].

Many substantial research studies have been conducted on *Z. rhetsa* root bark, but none has examined its anti-diabetic effects. Aarti N *et al.*, carried out a similar investigation using *Eulophia ochreata* L.'s *in vitro* antioxidant, antiglycation, and alpha-amylase inhibitory capabilities [27]. According to Aliyar MA *et al.*, studies, the seed extract of *Syzygium cumini* (L.) has insulin secretagogue action by *in vitro* anti-diabetic activity, bioactive components, and molecular modeling investigations using sulfonylurea receptor 1 protein [28]. Zhang H *et al.*, discovered a link between the genotype of sulfonylurea receptor 1 protein and the type 2 diabetes treatment responses to gliclazide [29]. Abbasi A *et al.*, indicate that human peroxiredoxin family proteins play a critical role in the overt hyperglycemia that results from Type 2 Diabetes pathogenesis [30]. Moreover, the human peroxiredoxin 5 and alpha-amylase receptor proteins are also involved in oxidative stress and it is evident from some previous research works conducted by Alminderej F *et al.*, 2020, Raju L *et al.*, 2022, and Pacifici F *et al.*, 2014 [31–33]. Furthermore, despite prior *in vitro* antioxidant investigations [20] employing the root bark of *Zanthoxylum rhetsa* plant, no *in vitro* and *in silico* study has yet been carried out against our reported three proteins.

The current research investigated the anti-diabetic and antioxidant properties of the *Z. rhetsa* bark root through *in vivo* and *in vitro* investigation. In addition, molecular docking studies, ADMET characteristics as well as MD simulation experiments were conducted to support *in vitro* and *in vivo* results by examining the function of reported phytocompounds selected by GCMS profiling.

## Materials and methods

### Ethics statement

All animal-based investigations were conducted by following the ethical guidelines adopted by the Animal Ethics Committee of R. P. Shaha University (Protocol number-RPSU/Registrar/ECR/Phr/2020/46). Mice were orally given a 2% glucose solution to prevent hypoglycemia following an alloxan injection. Upon completion of the studies, they were euthanized using diethyl ether anesthesia, with all attempts made to minimize any distress or pain.

## Plant collection and crude sample preparation

Collected root barks of *Z. rhetsa* (Tangail, Bangladesh) were identified by botany experts in the National Herbarium Centre in Bangladesh (DACB 43736). The root barks were washed with tap water. Then, it was allowed to shed drying for 15 days. The dried root barks were crushed with a grinder to make coarse powder. About 438.52 g of powder was macerated with 95% ethanol in an amber glass container for 15 days. To remove the ethanol from the mixture and to obtain the crude extract for this study, it was filtered and rotary evaporated at 50˚C.

## Experimental animals

Young, healthy Swiss-albino mice (4–5 weeks old, 18–25 g weight each) of both sexes were collected from Jahangirnagar University, Bangladesh. Animals were maintained in standard living environments at $(25\pm1)$˚C temperature, 56%-60% relative humidity and standard rodent food and water were made available for them for 1 week for proper adaptation/acclimatization. Mice were carefully handled, supplied food and water regularly, and measured body weight twice a week.

## Chemicals and reagents

Acetone, ascorbic acid, aluminum chloride, chloroform, dibasic phosphate, DPPH, DMSO, ethanol, ferric chloride, gallic acid, hydrochloric acid, iodine, methanol, n-hexane, potassium ferricyanide, quercetin, sodium carbonate, sodium monobasic phosphate etc. were of standard grade and were purchased from Mark, Germany. Alloxan, Folin–Ciocalteu (FC), *p*-nitrophenyl-α-D-glucopyranoside (pNPG), α-amylase, α-glucosidase were taken from Sigma ltd., St. Louis, USA. Acarbose and glibenclamide were obtained from Pacific Pharma, Bangladesh.

## Acute toxicity study

The acute toxicity test was accomplished on albino mice to find out any adverse effects of plant extract. Different doses such as 0.5, 1.0, 2.0, and 3.0 g per kg body weight (bw) of ZRRB were administrated orally, while mice in the control group were given 10% Dimethyl sulfoxide (DMSO) solution. After the extract administration, experimental mice were observed for 7 days to notice the different behavior of animals such as convulsion, writhing reflex, movement, tremor, salivation, urination, morbidity, and mortality were observed [34].

## Analysis of antioxidant property

**Qualitative antioxidant activity test.** The qualitative antioxidant property was studied via the Thin-layer chromatography (TLC) method. The TLC plates were developed using nonpolar, medium polar, and polar solvent systems. Colour changes were observed by spraying a 0.02% DPPH solution on TLC plates [34].

**Quantitative antioxidant activity test.** *DPPH free radical assay*. By using a standard protocol, the ability of ZRRB extract to neutralize DPPH radicals was measured. Briefly, 3 mL of a newly made DPPH solution (0.004% w/v) and 1 mL of sample was mixed and the absorbance was noted at 517 nm [34].

% Inhibition was counted by the following equation.

$$\% \ inhibition = \left(1 - \frac{Absorbance \ of \ sample}{Absorbance \ of \ the \ blank}\right) \times 100 \tag{1}$$

### Reducing power assay

An established approach [34] was applied here. Briefly, an aliquot (1 mL) of extract was taken and then mixed with 2.5 mL of phosphate buffer, 2.5 mL of potassium ferricyanide, and 2.5 mL of trichloroacetic acid. Sample absorbance was measured at 700 nm.

### Measurement of total antioxidant capacity (TAC)

To measure the TAC of the ZRRB extract, a mixed reagent was made by $H_2SO_4$ (0.6 M), $Na_3PO_4$ (28 mM) and ammonium molybdate (4 mM) following the 4: 2: 4 ratios. Subsequently, 300 μL of extract and ascorbic acid of different concentrations (200–6.25 μg/mL) was dissolved reagent mixer (3 mL) and allowed for incubation (95˚C for 90 min). The absorbance of the ZRRB extract was recorded at 695 nm. The TAC was shown as ascorbic acid equivalents per gram of dry extract (mg AAE/g) [35].

### Measurement of the total phenolic content (TPC)

In accordance with the described method [35], the TPC of the *Z. rhetsa* extract was calculated using the Folin–Ciocalteu (FC) reagent. Briefly, a sample (1 mL) was combined with distilled water (9 mL) followed by one mL (1 mL) of 10% v/v FC reagent and 10 mL of 7% sodium carbonate solution was taken to the test tube. The TPC of the extract was expressed in mg gallic acid equivalent per gram.

### Measurement of the total flavonoid content (TFC)

The TFC of *Z. rhetsa* was also calculated using the aluminum chloride calorimetric technique [35]. To carry out this experiment, a sample (1 mL) was mixed with sodium nitrous solution (5% w/v, 0.3 mL), and distilled water (4 mL), allowing the mixer to react (5 min). The TFC of ZRRB was calculated as mg QE/g of dry extract.

### Measurement of the total tannin content (TTC)

The procedure for calculating TTC was previously described [35]. TTC was estimated by constructing a calibration curve. After adding 7.5 mL distilled water and FC reagent (0.5 mL) to the ethanol extract (0.1 ml), 1 mL of 35% sodium carbonate solution was mixed. The absorbance of the sample was taken at 725 nm after the reaction (0.5 H) at room temperature.

### Evaluation of anti-diabetic activity

**Oral glucose tolerance test (OGTT).** To conduct OGTT, mice were kept without meals for 16 hours before the investigation. The experimental animals were divided into following groups: control group (Mice treated with 1% tween 80 in water at a dose of 10 mg/kg bw), positive control group (Mice administered glibenclamide at a dose of 5 mg/kg b.w.), and test groups (One group mice received ZRRB of 250 mg/kg bw, whereas other groups received ZRRB at a dose of 500 mg/kg). Collection of blood was done from the tail vein of mice to check the sugar level at 0, 30, 60, 90, 120, and 150 minutes [34].

**Alloxan-induced anti-diabetic test.** The mice were kept overnight fasting but only with water ad libitum. The mice were given freshly made alloxan solution (in normal saline) intraperitoneally only a single dose (150 mg per kg of body weight) to develop diabetes. Blood collection was collected after 10 days of alloxan solution and using an electronic glucometer fasting blood sugar (FBS) was measured the sugar level of mice higher than 10 mmol/L was considered as diabetic and selected for this test. Finally, mice with diabetes were arranged into five groups containing six animals. Group1 (Control): Mice were given saline orally; Group 2

(Diabetic control): Mice administered alloxan solution intraperitoneally; Group 3 (Positive control): Diabetic + Mice received the standard drug glibenclamide (5 mg/kg bw); Group 4 (Lower dose ZRRB): Diabetic + ZRRB dose (250 mg/kg bw); Group 5 (Higher dose ZRRB): Diabetic + ZRRB dose (500 mg/kg bw). The blood sugar levels of the animals were recorded at 10, 17, 24, and 31 days [34].

**α-Amylase inhibitory activity assay.** The starch-iodine method was employed to determine the inhibitory activity of α-amylase by ZRRB [34]. Briefly, 1 ml of sample, α-amylase solution (20 μL, 2 units/ mL), and PBS (1 mL) were taken and incubated at 37˚C for 10 min. The test tubes were then re-incubated for another 60 minutes with a freshly produced starch solution (1%, 200 L). Then, 10 mL of distilled water and 200 L of 1% iodine solution (containing 5 mM $I_2$ and 5 mM KI) were added to each test tube. After that, we utilized UV spectrophotometry at 610 nm to determine the absorbance.

The inhibitory activity (α-amylase) was determined using;

$$\% \; inhibition \; activity = \frac{Sample \; Absorbance - Blank \; Absorbance (Without \; enzyme)}{Blank \; Absorbance (Without \; enzyme) - Control \; Absorbance} \times 100 \; (2)$$

**α-Glucosidase inhibitory assay.** This extract was tested for its capability to suppress the action of the α-glucosidase using the previously established method [36]. Briefly, the mixture was prepared with 112 μL PBS (pH 6.8), α-glucosidase solution (20 μL, 1 unit/ mL), and 10 μL of the sample. After 15 minutes of reaction (37˚C), 20 μL of pNPG (2.5 mmol/L) was mixed in the test tube. Furthermore, the reaction was done at 37˚C for 15 min and the reaction was stopped using 80 μL of $Na_2CO3$ solution (0.2 mol/L). Sample absorbance was measured using UV-spectrophotometry at 405 nm, and acarbose was considered as a reference sample.

The enzyme inhibitory action was determined by;

$$\% \; \alpha-Glucosidase \; inhibition = \frac{Absorbance \; of \; sample - Absorbance \; of \; sample \; blank}{Absorbance \; of \; control - Absorbance \; of \; blank} \times 100 \quad (3)$$

## Identification of phytoconstituents using GCMS

The Clarus 690 Gas Chromatograph (PerkinElmer, CA, MA, USA) was used as an instrument to determine the phytoconstituents. Helium, a carrier gas, was used to insert the sample into the column and maintained a constant rate of 1 ml/min. The mass range of the Gas Chromatograph was set to 50–600 m/z, which also fixed the scan time (1 s). The inlet temperature was maintained at 280˚C. The phytocompounds of the extract were identified by comparing them with the National Institute of Standards and Technology (NIST) database [35].

## *In silico* prediction

**Studies of molecular docking.** *Protein preparations*. Antidiabetic activity of the ZR plant was studied by targeting three proteins including human peroxiredoxin 5 (PDB ID: 1HD2), sulfonylurea receptor 1 (PDB ID: 5YW7) and human α-amylase (PDB ID: 1HNY). Then the 3D structure of the reported three proteins was explored from the RCSB Protein Data Bank (http://www.rcsb.org/) [37]. Water molecules, existing inhibitors, and other heteroatoms were separated from protein 3D structures via Biovia Discovery Studio Visualizer [38]. Finally, energy minimization of each protein structure was completed using the GROMOS 43B1 force field of Swiss PDB Viewer (version 4.1.0) software [39].

*Ligand preparations*. The PubChem (https://pubchem.ncbi.nlm.nih.gov/) database contains 3D structures for all small molecules utilized in virtual screening [40]. Open Babel software was utilized to convert 2D SDF to the 3D SDF structure of all reported compounds in this

study [40]. After that, the Open Babel interface in PyRx was used to transform the 3D SDF to 3D PDB format for reducing negative free energy of each compound [41].

**Modelling and displaying molecular interactions.**   The molecular docking investigations were undertaken using the AutoDock Vina interface of the PyRx program. The recovered compounds were bound with the reported three proteins for further evaluation of drug-like compound evaluation [41]. A grid box was generated with dimensions X: Y: Z = 37.0337: 36:5568: 44:0776, 55.1733: 72.7916: 54.2851 and 42.7645: 47.3425: 61.5464 for 1HD2, 1HNY and 5YW7 PDB structures, respectively. To improve the outcome, blind docking rather than site-specific docking to place each compound in its proper place in the protein binding pocket was used. Each hit generated 9 conformations, with the top-ranked inhibitor conformations selected for advanced results. The patterns of hydrogen and hydrophobic bonds in the ligand/ protein combination were represented using Biovia Discovery Studio Visualizer (version 21) [38]. Finally, molecular dynamics simulation studies verified molecular-coupling results.

**Pharmacokinetics and drug-likeness properties analysis.**   For a chemical to be considered for use in the pharmaceutical industry, it must first pass extensive testing to establish its absorption, distribution, metabolism, excretion and toxicity (ADMET) properties. The online server pkCSM (https://biosig.lab.uq.edu.au/pkcsm/prediction) was used to derive the ADMET characteristics of the reported compounds [42]. Pharmacodynamic and drug-likeness properties including Lipinski rule of five, lipophilicity, water solubility, and physicochemical of reported compounds were additionally generated using a SwissADME web server (http:// www.swissadme.ch/)) [43]. Lipinski's rule of five was commonly employed to determine whether or not a substance had the potential to be a drug and compounds that adhered to the rule were viewed as promising drug candidates [44].

**Molecular Dynamics Simulation (MDS) studies.**   MDS was used at 100 ns to determine the interaction solidity of protein-ligand complex structures. Then Schrödinger's "Desmond v3.6 Program" (https://www.schrodinger.com/) (Paid version) was used within a Linux framework to execute MDS estimating several protein-ligand complex structures [45]. Throughout its MDS, the accuracy was measured by the simulation interaction diagram (SID) generated through the Desmond modules of this Schrödinger suite. Data for the Root means square deviation (RMSD), Root-mean-square fluctuation (RMSF), intramolecular hydrogen bonds, and radius of gyration (Rg) of this protein-ligand complex were used to evaluate the stability and flexibility of the complex.

## Statistical analysis

Data were calculated as mean ± SD and each group contained 6 mice. Analysis of the test results was done by unpaired *t*-test. GraphPad Prism version 8.0.2 was used to draw the figures [34].

## Results

### Acute toxicity studies

The body weight was not significantly altered throughout the experiment as shown in the figure (S1 Fig), as well as no other abnormalities and mortality were noticed.

### Antioxidant activity

**Qualitative antioxidant activity test.**   The qualitative antioxidative property was determined on a TLC plate by applying a DPPH solution. The extract showed light-yellow spots on the plate specifying the presence of antioxidative components.

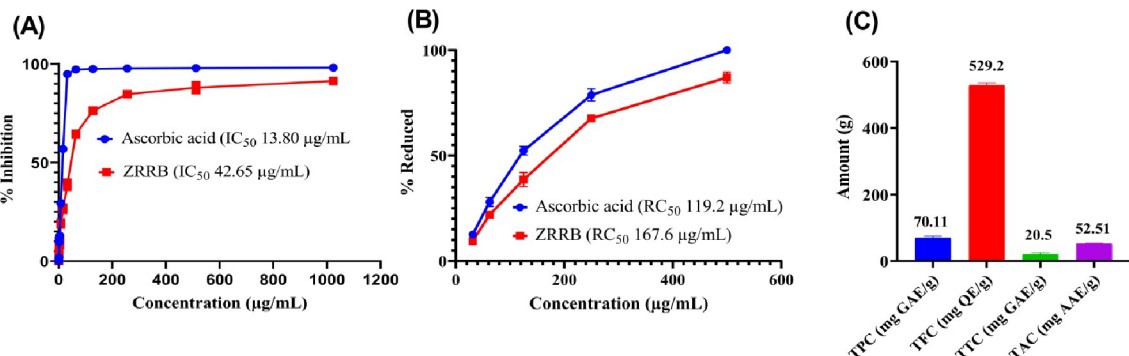

**Fig 1. Determination of the antioxidant activity of ZRRB extract.** (A) DPPH radical scavenging test; (B) reducing power assay; (C) TPC, TFC, and TTC as well as TAC of the extract.

**Quantitative antioxidant activity test.** The quantitative antioxidant activity test was measured by DPPH radical assay, and reducing power assay. The ZRRB extract exhibited notable antioxidant potentials in a concentration-dependent manner Fig 1A. We found that the $IC_{50}$ value (42.65 μg/mL) of the ethanolic extract of ZRRB was comparable with reference, ascorbic acid ($IC_{50}$ 13.80 μg/mL) (Fig 1A). In the reducing test, the $RC_{50}$ value of the ZRRB extract and positive control of ascorbic acid was calculated and was found as 167.6 μg/mL and 119.2 μg/mL, respectively (Fig 1B). In addition, we determined that this extract has a total antioxidant activity of 52.51 mg AAE/g after measuring its antioxidant capacity (Fig 1C).

## Secondary metabolites

We measured total secondary metabolite contents including phenolics, flavonoids and tannins using a standard calibration curve (S2 Fig). The TPC and TTC of ZRRB were found to be 70.11 GAE/g and 20.5 mg GAE/g of dry extract, whereas the TFC was 529.2 QE/g of extract (Fig 1C).

## Antidiabetic activity

**Oral glucose tolerance test.** In this evaluation, all mice blood sugar levels were highly elevated compared to fasting blood glucose concentrations at 0 min (range of blood glucose levels at 0 min and 30 min: 4.5–6.50 and 17.5–23 mmol/L). Our data also exhibited that both doses of ZRRB extract significantly lowered glucose levels in the blood ($p < 0.05$) at 60, 90, 120, and 150 minutes in Fig 2A compared to the control. Fig 2A shows that extract at higher doses lowered the blood sugar level more than 1.5 folds at 150 minutes (Control vs. 500 mg/kg dose: 11.01 mmol/L vs. 6.56 mmol/L).

**Alloxan-induced antidiabetic test.** In alloxan induced diabetic model, animals that received only an alloxan solution showed a 3-fold higher fasting sugar level and high glucose level was retained for more than 3 weeks compared to untreated mice, clearly indicating a diabetic animal model by alloxan introduction. As shown in Fig 2B, the extract of ZRRB (250 mg per kg and 500 mg per kg) significantly ($P < 0.05$) lowered glucose in the blood on days 24 and 31 compared to the diabetic control group.

**α-Amylase inhibitory activity assay.** In α-amylase inhibitory test, it appeared that the inhibitory activity of the enzyme by the extract was concentration-dependent. The result showed that the $IC_{50}$ value of the ZRRB extract was 81.45 μg/mL, whereas the $IC_{50}$ value of the standard drug acarbose was 14.05 μg/mL (Fig 2C).

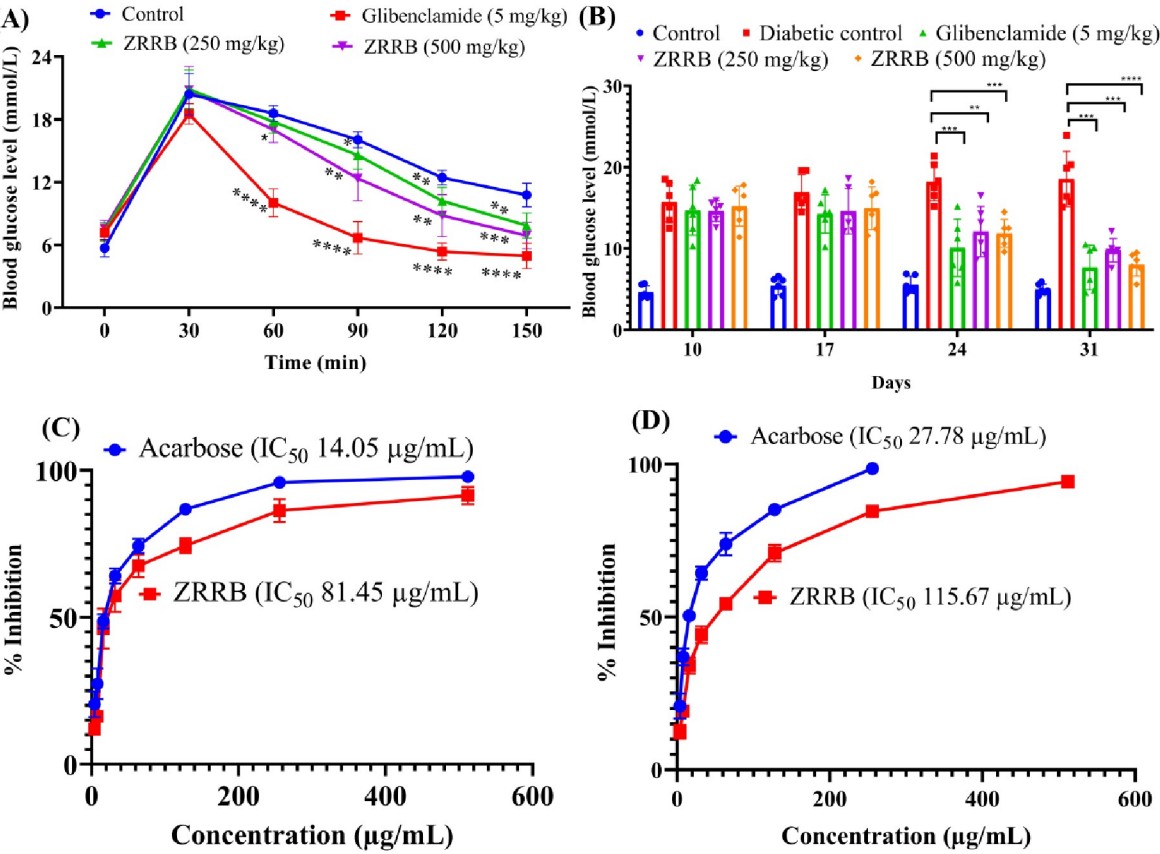

**Fig 2. Evaluation of the antidiabetic activity of ZRRB extract.** (A) Average blood glucose level at different time points after administration of sample/extract in OGTT test; (B) Average fasting blood glucose level at different days interval after administration; (C) Concentration vs % inhibition curve for α-amylase enzyme inhibitory activity test; (D) Concentration vs % inhibition curve for α-glucosidase inhibitory activity assay.

**α-Glucosidase inhibitory activity assay.** In α-glucosidase enzyme inhibition assay, it revealed that the extract had an IC$_{50}$ value of 115.67 μg/mL as illustrated in Fig 2D, whereas the IC$_{50}$ value of the standard drug acarbose was 27.78 μg/mL.

**GCMS analysis.** The present investigations of the GCMS analysis confirmed 16 different compounds. Among 16 different compounds, Dibutyl Phthalate showed the highest peak area (10.37%) in the chromatogram Table 1. The GCMS chromatogram of ZRRB is also shown in figure (S3 Fig).

## Molecular interaction analysis

To find a phytocompound that was effective against the human peroxiredoxin 5, human α-amylase and sulfonylurea receptor 1 (SUR 1) proteins, a total of 16 phytocompounds were docked (S1 Table) in this investigation. The interactions of the docked ligands were compared with that of standard drugs ascorbic acid (CID: 54670067) against human peroxiredoxin 5 for antioxidant potential, glibenclamide (CID: 3488) against SUR-1 protein and acarbose (CID: 41774) against human α-amylase proteins for antidiabetic potential, to evaluate the effectiveness of the interaction. The highest binding energy value compounds (-kcal/mol) and amino acids of the ligand binding site were measured after ligands with the selected protein interaction in Fig 3 and Table 2. In the determination of the antioxidant potential of the ZRRB extract, the result showed that standard drug ascorbic acid had -5.6 kcal/mol binding affinity

**Table 1. Identification of phytocompounds by GCMS technique.**

| S.NO | RT | Peak area % | Ligands | Formula | PubChem ID | MW |
|---|---|---|---|---|---|---|
| 1 | 6.03 | 1.20 | 3-(1'-pyrrolidinyl)-2-butanone | $C_8H_{15}NO$ | 23388489 | 141.21 |
| 2 | 7.42 | 5.56 | Dl-threitol | $C_4H_{10}O_4$ | 8998 | 122.12 |
| 3 | 9.09 | 0.73 | Trans-2,4-dimethylthiane, s,s-dioxide | $C_7H_{14}O_2S$ | 3016713 | 162.1 |
| 4 | 14.15 | 0.72 | Chloroacetic acid, tetradecyl ester | $C_{16}H_{31}ClO_2$ | 536326 | 290.9 |
| 5 | 20.06 | 0.70 | 2,4-difluorobenzoic acid, 2-propylphenyl ester | $C_{16}H_{14}F_2O_2$ | 91717595 | 276.28 |
| 6 | 23.08 | 0.97 | 3',5'-dimethoxyacetophenone | $C_{10}H_{12}O_3$ | 95997 | 180.2 |
| 7 | 24.75 | 2.89 | Beta.-l-arabinopyranoside, methyl | $C_6H_{12}O_5$ | 102169 | 164.16 |
| 8 | 27.32 | 2.15 | Methyl 11-methyl-dodecanoate | $C_{14}H_{28}O_2$ | 4065233 | 228.37 |
| 9 | 27.49 | 0.90 | (E)-4-(3-hydroxyprop-1-en-1-yl)-2-methoxyphenol | $C_{10}H_{12}O_3$ | 1549095 | 180.2 |
| 10 | 30.80 | 10.37 | Dibutyl phthalate | $C_{16}H_{22}O_4$ | 3026 | 278.34 |
| 11 | 30.90 | 2.06 | 13-octadecenoic acid, methyl ester | $C_{19}H_{36}O_2$ | 5364506 | 296.49 |
| 12 | 31.11 | 1.55 | 12,15-octadecadienoic acid, methyl ester | $C_{19}H_{34}O_2$ | 5365571 | 294.5 |
| 13 | 31.53 | 1.84 | Methyl 8,11,14-heptadecatrienoate | $C_{18}H_{30}O_2$ | 85978449 | 278.4 |
| 14 | 34.08 | 2.04 | 2H-1-benzopyran-2-one, 3,4,7-trimethoxy | $C_{12}H_{12}O_5$ | 606454 | 236.22 |
| 15 | 36.52 | 3.78 | Hentriacontane | $C_{31}H_{64}$ | 12410 | 436.8 |
| 16 | 39.17 | 7.02 | 6H-indolo[3,2,1-de][1,5]naphthyridin-6-one | $C_{14}H_8N_2O$ | 97176 | 220.23 |

with VAL94, LEU96, GLU16, GLU91, ALA90, LEU96, GLY82, and ARG86 amino acid residues (Fig 4A). Among 16 compounds, two phytocompounds such as 6H-indolo[3,2,1-de][1,5] naphthyridin-6-one (CID: 97176) and 2H-1-benzopyran-2-one, 3,4,7-trimethoxy (CID: 606454) demonstrated higher binding energy than the standard drug in human peroxiredoxin 5 (Table 2). The highest binding compound CID: 97176 interacts with GLU16, ARG86, GLY82, LEU96, ALA90, and VAL94 residues in the catalytic site of the protein via hydrogen bond formation (Fig 4A and Table 2). For human α-amylase protein, only two

6H-indolo[3,2,1-de][1,5]naphthyridin-6-one
(CID: 97176)

(E)-4-(3-hydroxyprop-1-en-1-yl)-2-methoxyphenol
(CID: 1549095)

2,4-difluorobenzoic acid, 2-propylphenyl ester
(CID: 91717595)

2H-1-benzopyran-2-one,3,4,7-trimethoxy
(CID: 606454)

**Fig 3. Structure of compounds identified GCMS analysis that showed highest binding affinities for human peroxiredoxin 5, α-Amylase, and sulfonylurea receptor 1 proteins.**

**Table 2. Interaction of ligands showing high docking scores for human peroxiredoxin 5, α-amylase and sulfonylurea receptor 1.**

| Target Enzyme (PDB ID) | Ligands (CID) | Binding affinity (kcal/mol) | Interacting amino acid |
|---|---|---|---|
| Human peroxiredoxin 5 (1HD2) | Ascorbic acid (54670067) | -5.6 | VAL94, LEU96, GLU16, GLU91 |
| | (E)-4-(3-hydroxyprop-1-en-1-yl)-2-methoxyphenol (1549095) | -5.3 | ALA90, LEU96, GLY82, ARG86 |
| | 2,4-difluorobenzoic acid, 2-propylphenyl ester (91717595) | -6.1 | GLU16, ARG86, GLU91, GLY85, ALA90, LEU96, GLY17 |
| | 2H-1-benzopyran-2-one, 3,4,7-trimethoxy (606454) | -5.7 | GLU91, ARG86, ASN21, GLU16 |
| | 3-(1'-pyrrolidinyl)-2-butanone (23388489) | -4.2 | LEU96, ARG86 |
| | 3',5'-dimethoxyacetophenone (95997) | -5.2 | ARG86, LEU96, ARG95, ALA90 |
| | 6H-indolo[3,2,1-de][1,5]naphthyridin-6-one (97176) | **-6.8** | GLU16, ARG86, GLY82, LEU96, ALA90, VAL94 |
| | Beta.-l-arabinopyranoside, methyl (102169) | -5 | GLU16, VAL94, GLU91 |
| | Dl-threitol (8998) | -4.2 | LEU96, VAL94, GLU16, GLY82, ARG86 |
| Human α-amylase (1HNY) | Acarbose (41774) | -7.7 | ARG421, PRO332, GLU282, LYS278, TRP280, SER289, ASP402 |
| | (E)-4-(3-hydroxyprop-1-en-1-yl)-2-methoxyphenol (1549095) | -6 | LEU162, ALA198, GLU233, ARG195, ARG197, TYR62 |
| | 2,4-difluorobenzoic acid, 2-propylphenyl ester (91717595) | **-7.4** | ASP300, ASP197, TYR62, LEU165, TRP59, GLN63 |
| | 2H-1-benzopyran-2-one, 3,4,7-trimethoxy (606454) | -5.9 | ASP300, ASP197, HIS101, TYR62, LEU165, GLN63, TRP59 |
| | 3-(1'-pyrrolidinyl)-2-butanone (23388489) | -4.5 | TYR62, LEU165 |
| | 3',5'-dimethoxyacetophenone (95997) | -5.2 | SER289, ARG252, PHE335, PRO4, ARG398, THR11 |
| | 6H-indolo[3,2,1-de][1,5]naphthyridin-6-one (97176) | **-7.7** | ASP300, TYR62, ASP197, LEU162 |
| | Beta.-l-arabinopyranoside, methyl (102169) | -4.6 | GLY334, ASP402, PRO332 |
| | Dl-threitol (8998) | -4.4 | ASN301, ILE312, GLN302, ASP317, ARG267, ALA310 |
| Sulfonylurea receptor 1 (5YW7) | Glibenclamide (3488) | -8.5 | ARG1246, ILE381, TYR377, TRP430, LEU592, LEU434, ASN437, LEU1241 |
| | (E)-4-(3-hydroxyprop-1-en-1-yl)-2-methoxyphenol (1549095) | -5.9 | VAL884, ILE873, LEU877, LEU708, ARG878, ASP898, HIS896, LYS881, GLY706, ARG882 |
| | 2,4-difluorobenzoic acid, 2-propylphenyl ester (91717595) | -7.6 | ASN437, ALA380, PRO436, ARG306, TYR377 |
| | 2H-1-benzopyran-2-one, 3,4,7-trimethoxy (606454) | -5.9 | LEU1241, THR1242, ILE381, ARG1246, ASN1245, MET429 |
| | 3-(1'-pyrrolidinyl)-2-butanone (23388489) | -5.1 | PHE433, PRO436, ALA380, ARG306, TYR377, LEU434 |
| | 3',5'-dimethoxyacetophenone (95997) | -5.7 | ARG1246, TYR377, LEU592, PRO436, ALA380 |
| | 6H-indolo[3,2,1-de][1,5]naphthyridin-6-one (97176) | **-8.1** | PRO436, ALA380, ARG306, LEU434, LEU592 |
| | Beta.-l-arabinopyranoside, methyl (102169) | -4.9 | SER403, GLU729, THR404, SER405 |
| | Dl-threitol (8998) | -3.9 | ARG306, ASP310, GLN369, GLN444 |

phytocompounds (CID: 97176 and CID: 91717595) interacted with the target protein through hydrophobic and hydrogen bond formations including ASP300, ASP197, TYR62, LEU165, TRP59, and GLN63 amino acid residues at the binding site (Table 2 and Fig 4B). On the other hand, SUR-1 protein, the best compound (CID: 97176) was bound with PRO436, ALA380, ARG306, LEU434, and LEU592 residues out of 16 tested compounds which showed binding energy (-8.1 kcal/mol) close to the control drug (-8.5 kcal/mol) (Fig 4C and Table 2).

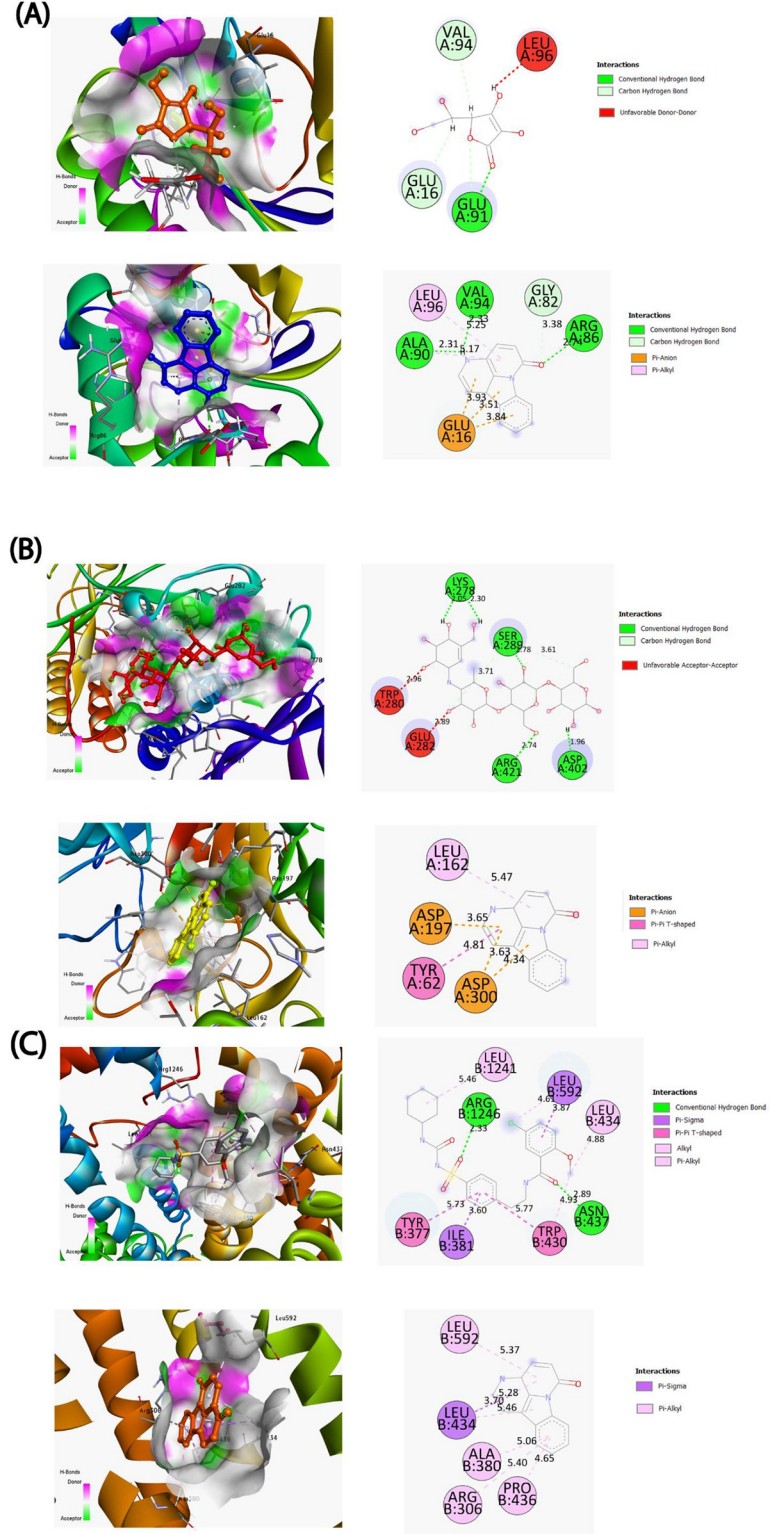

**Fig 4.** 3D and 2D molecular binding interaction of (A) human peroxiredoxin 5 (PDB: 1HD2) against phytocompound CID: 97176 and control drug (ascorbic acid); (B) Human α-amylase (PDB: 1HNY) against phytocompound CID: 97176 and control drug (acarbose); (C) Sulfonylurea receptor 1 (PDB: 5YW7) against phytocompound CID: 97176 and standard drug (glibenclamide).

### *In silico* pharmacokinetics and drug-likeness properties analysis

Compounds that have high binding affinity (CID: 1549095; 91717595; 606454; 95997; and 97176) were undergone ADMET analysis which is crucial for elucidating the safety profile of the mentioned top-scoring docked ligands (S2 Table). The low frequency of adverse CNS effects seen with CID: 97176 are supported by its high gastrointestinal absorption and lack of BBB permeability, as shown in S2 Table. All the reported ligands did not inhibit the activity of CYP2D6 and CYP3A4 isoforms indicating that there is a strong chance the compounds will be metabolized, act at the target sites, and then be eliminated. Besides, the levels of clearance for all the reported drugs were within acceptable ranges (S2 Table). Most of the reported compounds followed Lipinski's rule of five and fit these guidelines except CID: 91717595 showing only one violation. The mutagenicity of a chemical can be determined with the help of the Ames test. Both chemicals (except CID: 97176) were found to be non-Ames hazardous, suggesting that their carcinogenicity is debatable (S2 Table).

### MDS studies

MDS experiment was further employed to validate the molecular docking outcomes. To explain MDS findings, the RMSD, RMSF, Rg, and Intramolecular hydrogen bonding (Intra HB) have been used. By measuring the RMSD of a protein-ligand complex system, the mean distance produced by a selected atom's displacement over a given time period may be calculated. The protein human pancreatic α-amylase (PDB ID: 1HNY) was compared to the RMSD of the drug candidate compound CID: 97176 (blue) and the standard drug (Acarbose) (orange) complex structure, as shown in Fig 5A. In comparison to the structure of the natural protein, the RMSD for two compounds was within an acceptable range of 1.0 to 3.5 Å. Similarly, in Fig 5B the RMSD of drug candidate compound CID: 97176 (blue), and the standard drug (Glibenclamide) (orange) with the protein sulfonylurea receptor 1 (PDB ID: 5YW7) was calculated. Results showed that RMSD values ranged from 1 to 5 Å for both the candidate compound CID: 97176 and the control drug (Glibenclamide). Therefore, the RMSF values of the reported compound CID: 97176 (blue), and the standard drug (Acarbose) (orange) in complex with α-amylase (PDB ID: 1HNY) were considered to examine how attaching specific ligand molecules to a residual site impacts protein structural flexibility, as shown in Fig 6A. Additionally, the RMSF values of the selected compound CID: 97176 (blue), and the reference drug (Glibenclamide) (orange) with the sulfonylurea receptor 1 (PDB ID: 5YW7) are presented in Fig 6B. The possibility of a single-atom displacement fluctuating in the simulated environment is thus found to be small for the ligand complexes under this investigation, as illustrated in Fig 6A and 6B. As shown in Fig 7A and 7B, the mean Rg values of the selected compound CID: 97176 (blue), and the standard drugs (orange) were calculated at 2.9 Å and 6.0 Å for human pancreatic α-amylase while 2.9 Å and 5.4 Å for sulfonylurea receptor 1 proteins, demonstrating that protein-ligand interaction did not cause any significant structural changes to the binding site.

### Discussion

Phytochemicals evaluation expressed that the ZRRB extract contains glycoside, alkaloids, phenolics, tannins, flavonoids, saponins, and reducing sugar. Toxicity evaluation is an important marker for understanding the toxic index of extract. The body weight and behavior of the mice remained normal throughout the experiment. Thus, our results clearly express that the ZRRB extract is safe up to 3000 mg/kg and can be easily administered to the experimental animal. The antioxidant capability of the ZRRB extract was investigated both qualitatively and quantitatively in this investigation. Free radicals are generally formed due to normal metabolic

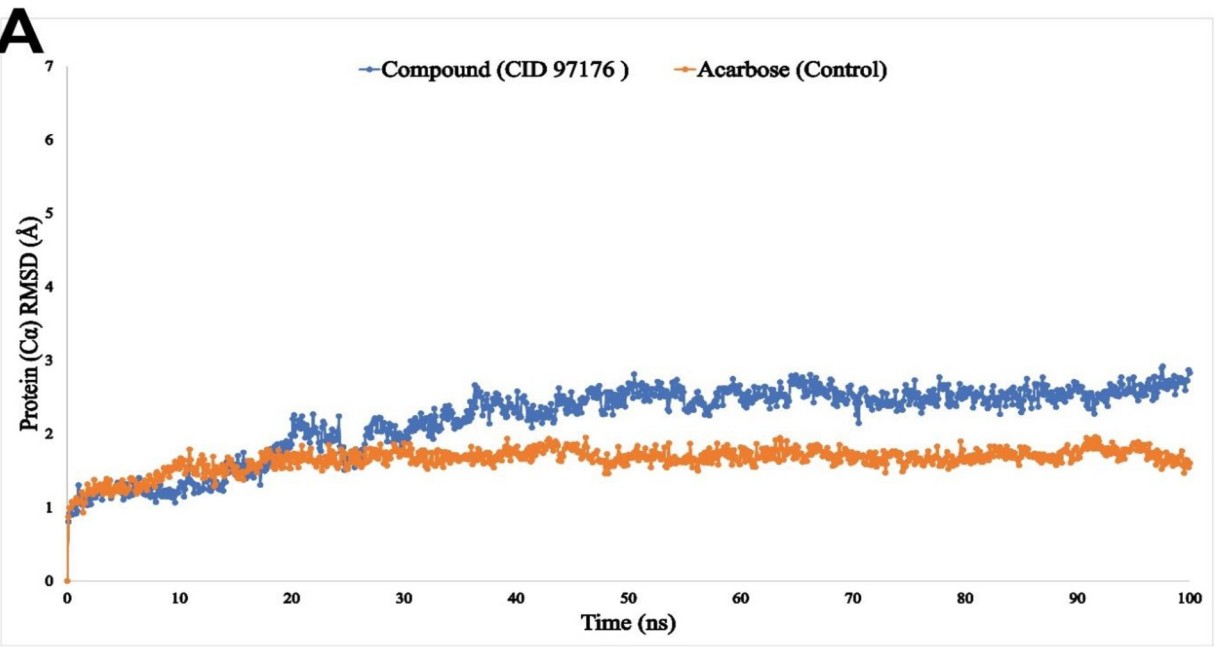

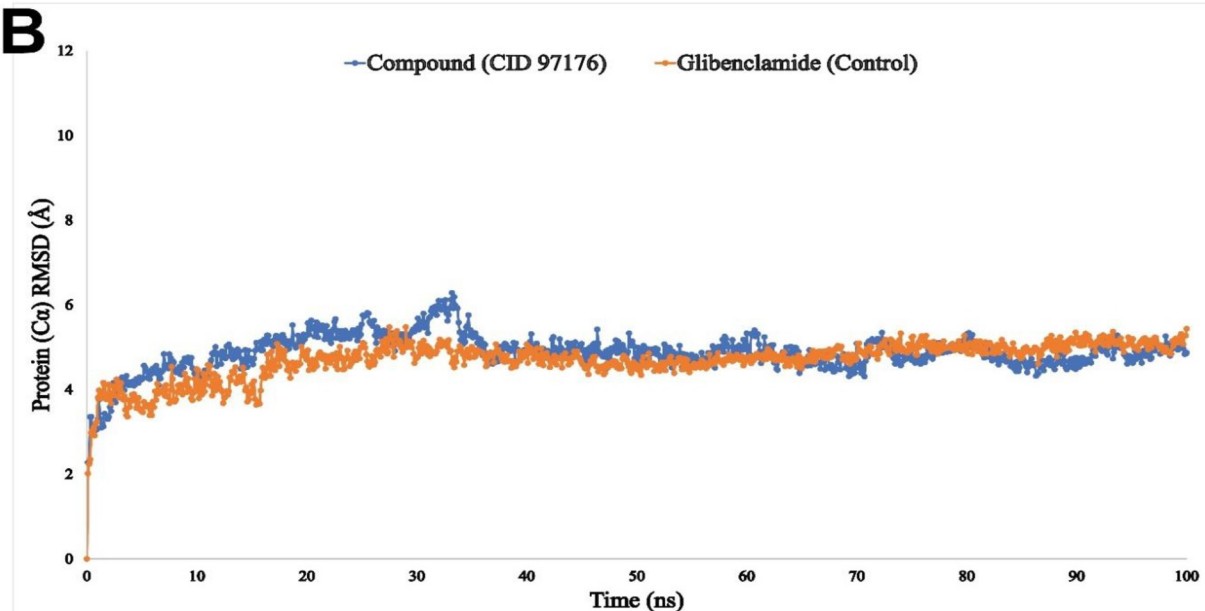

**Fig 5. The RMSD values of the two compounds bound to the two selected proteins are plotted. (A)** Human pancreatic **α**-amylase (PDB ID: 1HNY), where the selected ligand is CID: 97176, and the control drug (Acarbose); **(B)** Sulfonylurea receptor 1 where the selected ligand is CID: 97176, and the standard drug (Glibenclamide) in 100 ns MDS calculations. The selected ligand and control drugs indicate blue and orange color, respectively.

activities in the living body that are neutralized in healthy conditions through the activity of antioxidant molecules [46].

DPPH is commonly recognized as a free radical. The reaction of antioxidant compounds with stable DPPH solution turns it from deep violet to light yellow color [47]. In the reducing ability test, mechanistically, the antioxidant compounds lower the potassium ferricyanide

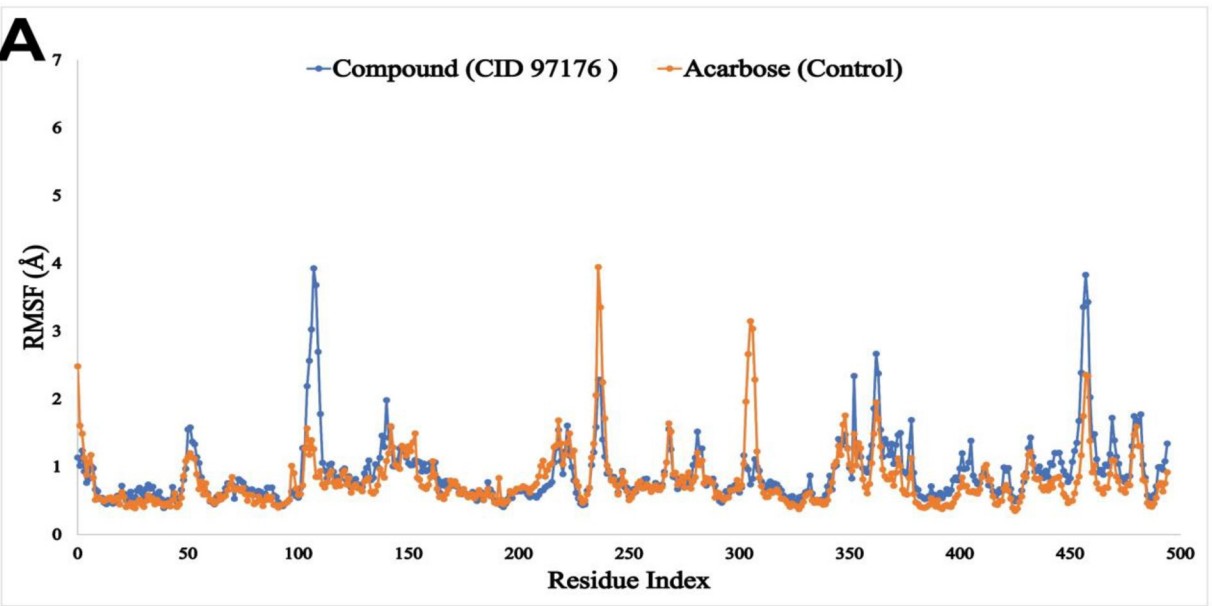

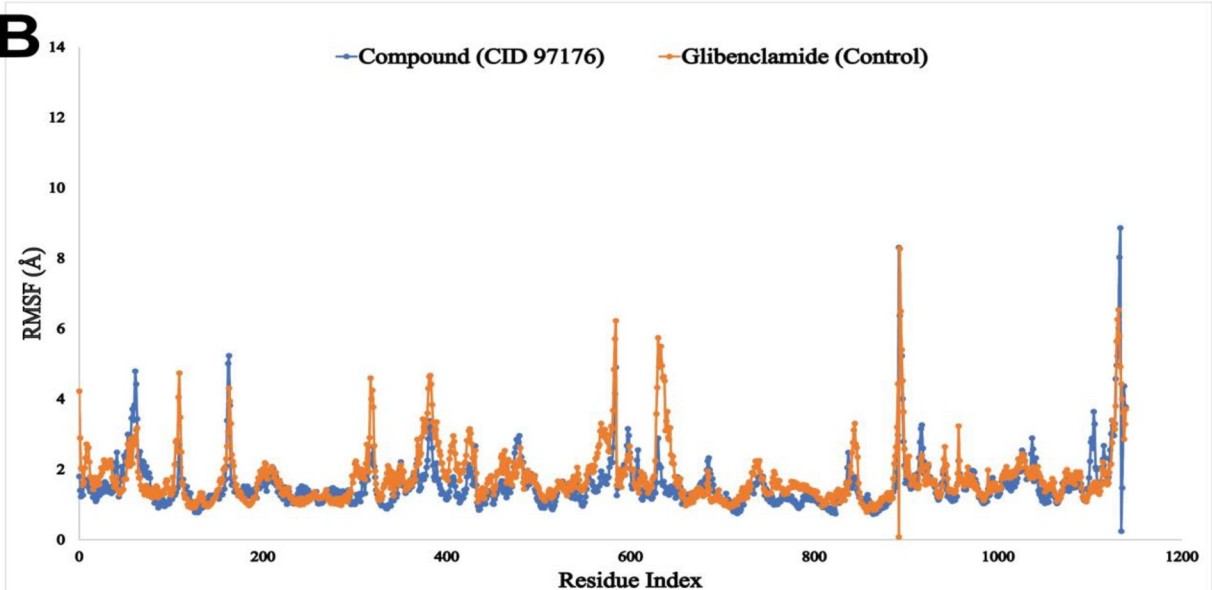

**Fig 6. The figures illustration the RMSF values for the two ligand molecules in complex with the targeted two proteins. (A)** Human pancreatic α-amylase (PDB ID: 1HNY) where the selected ligand is CID: 97176 (blue), and the control drug (Acarbose); **(B)** Sulfonylurea receptor 1 (PDB ID: 5YW7), where the selected ligand is CID: 97176, and the control drug (Glibenclamide) in 100 ns MDS calculations. The selected ligand and control drugs indicate blue and orange color, respectively.

($Fe3^+$) into potassium ferrocyanide and further insertion of ferric chloride ($FeCl_3$) develops Prussian blue [48]. Our data demonstrated a substantial amount of DPPH scavenging activity, ferric reducing power assay as well as total antioxidant capacity (Fig 1), highlighting its crucial antioxidant potential. Studies have demonstrated that polyphenols are common secondary metabolites that have antioxidant activities [49]. Ingestion of antioxidant-containing foods in daily diet can contribute to boost our immune systems which enhances cellular protection from free radicals-induced effects [50]. Based on the preliminary phytochemical assay and

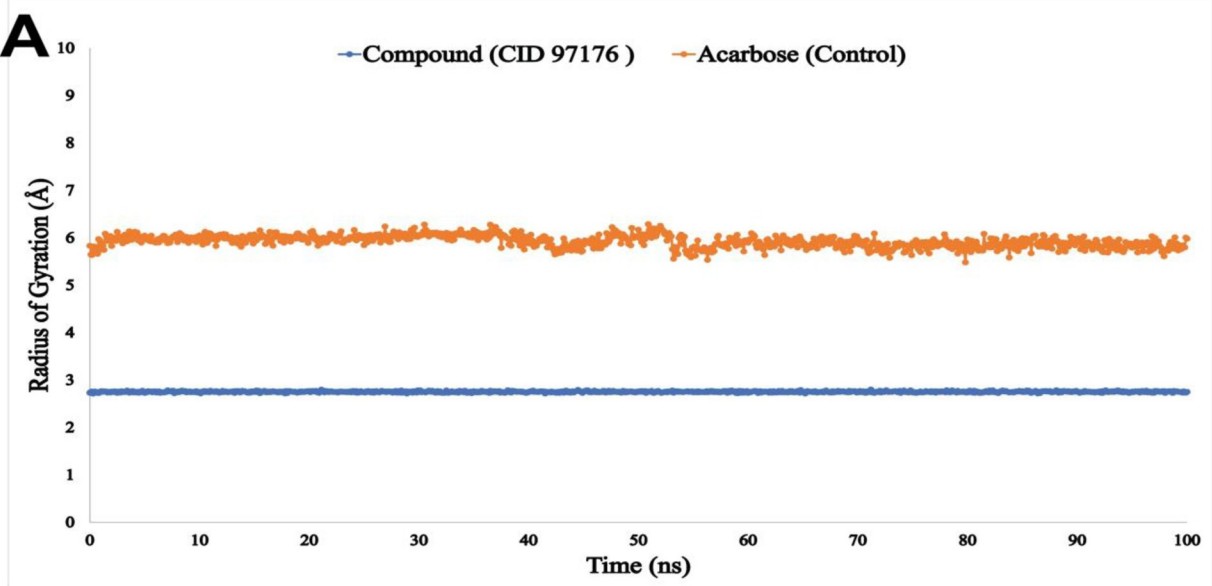

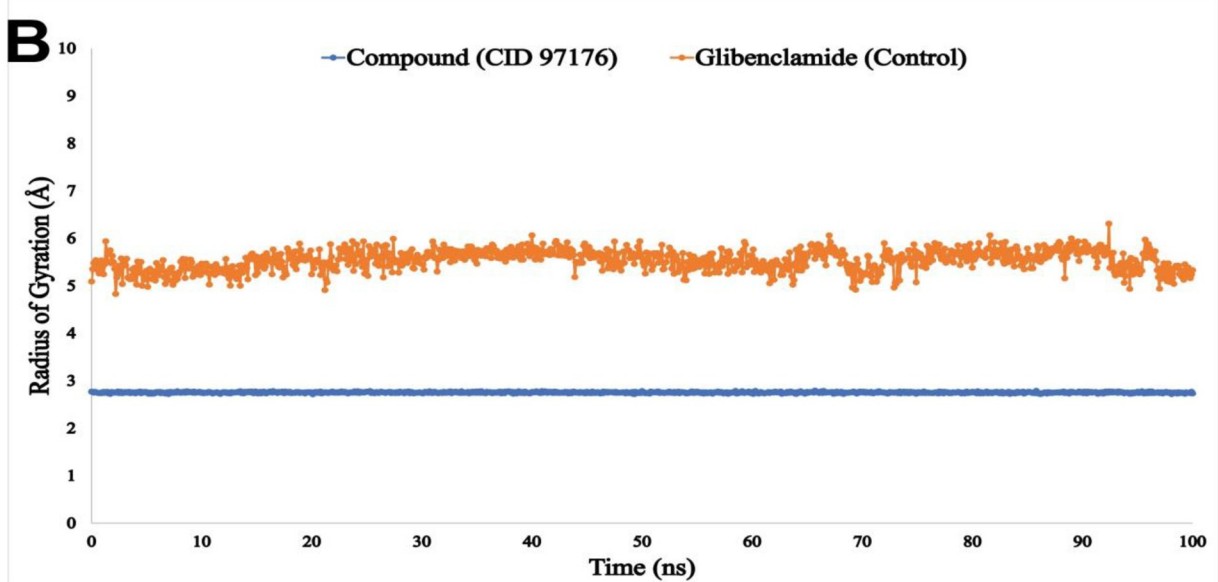

**Fig 7. The graphs depict the Rg values for the two ligand molecules bound to the two targeted proteins. (A)** Human pancreatic **α**-amylase (PDB ID: 1HNY), where the selected ligand is CID: 97176, and the control drug (Acarbose); **(B)** Sulfonylurea receptor 1 (PDB ID: 5YW7), where the selected ligands is CID: 97176 and the control drug (Glibenclamide) in 100 ns MDS assessments. The selected ligand and control drugs indicate blue and orange color, respectively.

GCMS analysis (Table 1), it is indicated that the ZRRB extract contains such compounds which may be responsible for such antioxidative behavior.

Flavonoids and phenolics are commonly known as secondary metabolites found in plants which have antioxidant properties. As plants are available in nature, research on plant-based antioxidant components is increased due to their capacity to defend living cells against oxidative stress-related disorders like diabetes [51]. Phenolic compounds contain aromatic rings with hydroxyl groups that are mainly responsible for antioxidant potential and also have

noncoupled electrons to neutralize free radicals [48]. We found that the ZRRB extract has a higher percentage of flavonoids and phenolics (Fig 1C) which may exert its antioxidative effects.

Phenolic, flavonoids, and tannins have the ability to control high blood glucose levels [52]. OGTT, alloxan induced antidiabetic mouse model, α-amylase, and α-glucosidase enzyme inhibition tests were done to investigate the antidiabetic activity of this extract. The OGTT test has been employed for many years to detect pre-diabetes and type 2 diabetes [53]. Our data also exhibited that both doses of ZRRB extract significantly lowered glucose levels in blood ($p < 0.05$) at different time intervals compared to the control (Fig 2A). Alloxan, a compound, is commonly employed in experiments to develop diabetes in mice. It is indicated that ZRRB extract significantly ($P < 0.05$) lowered sugar levels in comparison to the diabetic control group (Fig 2B). With regards to enzyme inhibition, we found that ZRRB extract was more effective against α-amylase than α-glucosidase (Fig 2C and 2D). α-amylase initiates the breakdown of carbohydrates by breaking down 1, 4-glycosidic bonds in polysaccharides (such as starch and glycogen) into disaccharides. α-glucosidase then catalyzes the conversion of the disaccharides into monosaccharides, resulting in postprandial hyperglycemia. Plants with high phenolic chemical content are thought to be a potential source of potent hypoglycemic drugs because they can prevent the enzymatic activity of α-amylase and α-glucosidase [47]. Previously reported that secondary metabolites such as phenolics, quercetin, rutin, catechin, and flavonoids are important phytochemicals that act as α-amylase and α-glucosidase inhibitory activity [54] by interaction with proteins that lead to the complex formation and changing conformation [55]. The observed antidiabetic activity by ZRRB extract is because it mainly contains various compounds like phenolics and flavonoids as compared to the pure inhibitor of acarbose drug.

Sixteen (16) major compounds were confirmed by GCMS analysis of the investigating extract and their percentage of peak area, PubChem CID is shown in Table 1. Plants with their compounds have the ability to reduce the elevated sugar level due to the existence of terpenoids, phenolic, flavonoid and tannin compounds [56]. It is documented that some natural compounds namely isorhamnetin [57], isovanillic acid [58], isoeugenol [59] and quercetin [60] have antioxidant and antidiabetic properties.

To explore the activity of compounds identified by GCMS analysis were further tested against selected proteins using molecular docking. Molecular docking is the best tool for quickly screening the protein-ligand interaction of an enormous number of natural compounds. High-affinity receptor-binding molecules may deliver a preliminary point for the creation of a novel therapeutic [61]. The energetic stability of a ligand at the protein active site is significantly enhanced by weak intramolecular interactions (hydrophobic and hydrogen bonds) and improved drug efficacy. Drugs may bind to their target more selectively and are less likely to have off-target effects if they can form specific hydrogen bonds. On the other hand, hydrophobic interactions promote target-drug interface binding affinity, biological activity, and biochemical stability among complex compounds [62]. Our molecular docking study suggests that 6H-indolo[3,2,1-de][1,5]naphthyridin-6-one (CID: 97176) compound revealed higher binding affinity against targeted proteins via forming several intramolecular interactions which are comparable to the standard drug, thus it can be an effective lead compound for both antioxidant and antidiabetic potential (Table 2).

There are undoubtedly very few chemical compounds that have either medicines or drug prospects. To eliminate substances with unwanted qualities, particularly those with poor ADMET, drug-likeness has been employed effectively [63]. Drug-likeness properties including Lipinski's rule of five (<5 hydrogen bond donors, <10 hydrogen bond acceptors, <500 dalton molecular weight, and < partition coefficient or 5 log p) were used to select ligands for a potential drug candidate [64]. In our study, the S2 Table reveals that the chosen

phytochemicals have adequate drug-likeness and physicochemical qualities to be therapeutic drugs. Results indicated that all reported compounds (except CID: 91717595) would not cause any liver damage. Our overall data suggest that the drug candidate followed violate Lipinski's rule of five, indicating its safe usage as a drug candidate.

Protein-ligand complex stability in different environments, such as the human body, can be determined through molecular dynamic modeling [65]. By measuring the RMSD of a protein-ligand complex system, values outside of the acceptable range suggest a substantial conformational change in the protein, while variations on the order of 1–5 or 0.1–0.5 nm are totally acceptable [66]. The RMSF value is essential for determining the flexibility and fluctuation of the residues. Higher RMSF values, which measure dispersion, suggest a less tightly packed protein-ligand combination [67]. The Rg calculation displays complicated compactness fluctuations during simulation time and is one of the most essential structural identifiers of a macromolecule. A high Rg number suggests that the chemicals are not tightly bound to the protein, while a low Rg value indicates good compactness [68]. The stability of proteins and molecules is significantly enhanced by interactive hydrogen bonding. Here in this study, the 100 ns simulation was carried out with defined ligands and examined their intermolecular interactions. For human alpha-amylase and CID: 97176, the intermolecular hydrogen bonds were found in Asp300 and Glu282 positions within 0.7 and 1.6 nm respectively (S4A Fig). In the case of sulfonylurea receptor 1 protein and drug candidate CID: 97176, the maximum number of hydrogen bonds was observed in the Arg1246 position within 1.00 nm distance (S4B Fig). Finally, the investigated molecules formed several hydrogens, hydrophobic, ionic, and water bridge bonds and retained them until the simulation ended, enabling persistent protein binding [69]. MDS analysis showed that the compound CID: 97176 disclosed promising RMSD, RMSF, Rg data, and the highest hydrogen bonding with human pancreatic α-amylase (PDB ID: 1HNY) and sulfonylurea receptor 1 (PDB ID: 5YW7) compared to the standard drug acarbose and glibenclamide, respectively.

Therefore, MDS concluded that the compound, 6H-indolo[3,2,1-de][1,5]naphthyridin-6-one (CID: 97176) bound to the active site of both human α-amylase and sulfonylurea receptor 1 at stable way, indicating the inhibition of these proteins which ultimately block the oxidative stress and facilitate the insulin production.

## Conclusion

The *Zanthoxylum rhetsa* root bark (ZRRB) extract contains phenolics and flavonoid-type compounds and was confirmed through GCMS analysis as well as qualitative phytochemical tests. The extract demonstrated significant antioxidant and antidiabetic potentials both in *in vivo* antidiabetic mice models and several *in vitro* experiments including molecular docking and MDS studies with different receptor proteins. After a series of experiments, CID: 97176 (identified as a major compound in this plant extract through GCMS analysis) was found to have the highest binding affinity with peroxiredoxin-5, α-amylase, and sulfonylurea receptor 1 proteins highlighting this compound as a promising candidate for further development as antioxidant and anti-diabetic drugs. Our studies suggest that the ZRRB extract could be traditionally used in the management of oxidative stress and/or diabetes and further investigated for isolation of pure drug lead which may open a source/idea to discover better antioxidant and antidiabetic drugs.

## Supporting information

**S1 Fig. Observation of mice body weight after administration of different doses of ZRRB extract.**
(TIF)

**S2 Fig.** Standard calibration curves for the calculation of (A) total antioxidant activity (TAA); (B) total phenolic contents (TPC); (C) total flavonoid contents (TFC); and (D) total tannin contents (TTC) of ZRRB extract.
(TIF)

**S3 Fig. GCMS chromatogram of ZRRB.**
(TIF)

**S4 Fig. The stacked bar charts reveal that intramolecular bond interactions between proteins and ligands were studied for the entire 100 ns of simulation time.** The figure expresses the interaction of ligands with the targeted two proteins. (A) $\alpha$-amylase (PDB ID: 1HNY), where the selected compound (CID: 97176) and control drug (Acarbose); (B) Sulfonylurea receptor 1 (PDB ID: 5YW7), where the selected compound (CID: 97176) and control drug (Glibenclamide). The selected ligand and control drugs indicate blue and orange color, respectively.
(TIF)

**S1 Table. Binding affinity (-kcal/mol) for all compounds.**
(PDF)

**S2 Table. ADMET, physicochemical, lipophilicity and drug-likeness properties analysis of reported compounds.**
(PDF)

## Acknowledgments

The authors would like to thank the institutions that made it possible to carry out this research: R. P. Shaha University, Jashore University of Science and Technology Lab, Khulna University, Centre for Advanced Bioinformatics and Artificial Intelligence Research at Islamic University, Kushtia, and Pharmaceutical Technology, University of Dhaka, Bangladesh.

## Author Contributions

**Conceptualization:** Apurba Kumar Barman.

**Formal analysis:** Rabindra Nath Acharyya, Marjana Alam.

**Investigation:** Apurba Kumar Barman, Md. Habibur Rahman.

**Methodology:** Sumaiya Mahadi, Md Arju Hossain, Rahima Begum.

**Resources:** Apurba Kumar Barman, A. S. M. Monjur Al Hossain.

**Software:** Md Arju Hossain, Rahima Begum, Md. Habibur Rahman.

**Supervision:** Apurba Kumar Barman, A. S. M. Monjur Al Hossain.

**Visualization:** Nripendra Nath Biswas.

**Writing – original draft:** Md Arju Hossain, Rahima Begum, Marjana Alam.

**Writing – review & editing:** Apurba Kumar Barman, Nripendra Nath Biswas, A. S. M. Monjur Al Hossain.

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
