## [Decision Letter · Decision Letter 0]

22 Mar 2024

PONE-D-24-02018Assessing antioxidant, antidiabetic potential and GCMS profiling of ethanolic root bark extract of Zanthoxylum rhetsa (Roxb.) DC: Supported by in vitro, in vivo and in silico molecular modelingPLOS ONE

Dear Dr. HOSSAIN,

Thank you for submitting your manuscript to PLOS ONE. After careful consideration, we feel that it has merit but does not fully meet PLOS ONE’s publication criteria as it currently stands. Therefore, we invite you to submit a revised version of the manuscript that addresses the points raised during the review process.

In addition, The literature review needs further work and the authors should adequately make proper literature review. 

Please make sure that the manuscript formatting, especially the references, follows the journal requirements. 

We look forward to receiving your revised manuscript.

Kind regards,

Taye Beyene Demissie, PhD

Academic Editor

PLOS ONE

Reviewers' comments:

Reviewer's Responses to Questions

**Comments to the Author**

1. Is the manuscript technically sound, and do the data support the conclusions?

Reviewer #1: Yes

Reviewer #2: Yes

2. Has the statistical analysis been performed appropriately and rigorously? 

Reviewer #1: Yes

Reviewer #2: Yes

3. Have the authors made all data underlying the findings in their manuscript fully available?

Reviewer #1: Yes

Reviewer #2: Yes

4. Is the manuscript presented in an intelligible fashion and written in standard English?

Reviewer #1: Yes

Reviewer #2: Yes

5. Review Comments to the Author

Reviewer #1: Comments

1) The work is very interesting and studied for pre-clinical aspects. It will be useful for further research in the development of clinical studies.

2) It would be better to mention as anti-diabetes in the keywords instead of other words.

3) In silico should be in italics. In some instances, it is missing. It has to be checked and correct them.

4) In vivo should be corrected into italics.

5) References should be rechecked again according to the journal guidelines.

Report

The manuscript is well written and finds significance through in vitro, in vivo and in silico studies. My decision is ‘minor comments or revisions’. Therefore, I would strongly recommend the manuscript to be published in the Plos One Journal, after the above comments are addressed.

Reviewer #2: 

1. What is the rationale behind the selection of target proteins for anti diabetic potential? The diabetes Meletus involves complex pathogenesis and so many molecular pathways.

2. In, introduction the authors mentioned that there are no in-silico studies related to the anti oxidant activity. According to my opinion the authors should have gone for any significant protein/gene related to anti oxidant potential.

6. PLOS authors have the option to publish the peer review history of their article (what does this mean?). If published, this will include your full peer review and any attached files.

Reviewer #1: No

Reviewer #2: No

---

## [Author Response · Author response to Decision Letter 0]

3 May 2024

Dear Editor and Reviewer, 

We would like to thank you for your time and effort in reading through our manuscript and for the constructive feedback. Please find below a detailed breakdown of your comments and suggestions. Please also find attached the updated manuscript with the amended sections and track change versions. 

Academic editor

We thank the editor very much for their kind comments with regard to our work. We appreciate all the points raised, which we address below:

Academic editor

#1. The literature review needs further work and the authors should adequately make proper literature review. 

Author Reply: Thank you for your suggestions. As per your concern, we have reviewed the relevant literatures in detail and included it in the introduction section of this revised manuscript (marked as track change).

#2. Please make sure that the manuscript formatting, especially the references, follows the journal requirements. 

 Author Reply: As per the journal reference guidelines, we have carefully formatted the references.

Reviewer 1:

We are very grateful for the kind and constructive comments of the reviewer. The points raised by the reviewer 1 are addressed below:

#1. The work is very interesting and studied for pre-clinical aspects. It will be useful for further research in the development of clinical studies.

Author Reply: Thanks a lot for your nice comments regarding further research.

#2. It would be better to mention as anti-diabetes in the keywords instead of other words. 

Author Reply: We have included anti-diabetes in the keywords (marked as track change).

#3. In silico should be in italics. In some instances, it is missing. It has to be checked and correct them. 

Author Reply: We have used In silico in italics in the manuscript as per your suggestion (marked as track change).

#4. In vivo should be corrected into italics. 

Author Reply: We have used In vivo in the manuscript as in italics.

#5. References should be rechecked again according to the journal guidelines. 

Author Reply: Thanks for your valuable comments. We have formatted the references according to the journal guidelines.

Reviewer 2:

We greatly appreciate the kind comments of the reviewer about the quality of our work. We address the points raised by the reviewer 2.

#1. What is the rationale behind the selection of target proteins for anti-diabetic potential? The diabetes Mellitus involves complex pathogenesis and so many molecular pathways.

Author Reply: Thank you for your query. We do agree that diabetes mellitus involves complex pathogenesis and so many molecular pathways but in a single article, it is not possible to investigate too many mechanistic pathways. The rationale behind the selection of target proteins for anti-diabetic potential typically involves several factors including pathophysiological relevance, druggability, clinical need, market potential and existing therapeutic targets.

Studies conducted by Pacifici F et al., and Abbasi A et al., indicate that human peroxiredoxin family proteins play a critical role in the overt hyperglycemia that results from Type 2 Diabetes pathogenesis. Zhang H et al., discovered a link between the genotype of sulfonylurea receptor 1 protein and the type 2 diabetes treatment response to gliclazide. Besides, numerous investigations have demonstrated that the phytochemical substances found in medicinal plants with alpha-amylase inhibitory activity may be useful in treating certain conditions and that protein may serve as a biomarker for type 2 diabetes diagnosis. From the literature search, we have found that our targeted three proteins may reveal many important insights about diabetes mellitus management. And that’s why the mentioned three receptor proteins were selected.

#2. In the introduction the authors mentioned that there are no in-silico studies related to the anti-oxidant activity. According to my opinion, the authors should have gone for any significant protein/gene related to anti-oxidant potential.

Author Reply: Thank you for your comment. Along with other significant proteins/genes (UCP2, RAGE, PPAR-γ and NRF2) related to anti-oxidant potential, our investigated peroxiredoxin 5 protein and alpha-amylase protein have a significant role in antioxidant potential. Previously, other researchers such as Raju L et al., Pacifici F et al., and Alminderej F et al., reported about the involvement of this protein with oxidative stress. This rationality has now been mentioned in the introduction part of the revised manuscript.

---

## [Editor Report · Decision Letter 1]

14 May 2024

Assessing antioxidant, antidiabetic potential and GCMS profiling of ethanolic root bark extract of Zanthoxylum rhetsa (Roxb.) DC: Supported by in vitro, in vivo and in silico molecular modeling

PONE-D-24-02018R1

Dear Dr. HOSSAIN,

We’re pleased to inform you that your manuscript has been judged scientifically suitable for publication and will be formally accepted for publication once it meets all outstanding technical requirements.

Kind regards,

Taye Beyene Demissie, PhD

Academic Editor

PLOS ONE
---

## [Editor Report · Acceptance letter]

9 Aug 2024

PONE-D-24-02018R1 

PLOS ONE

Dear Dr. Hossain, 

I'm pleased to inform you that your manuscript has been deemed suitable for publication in PLOS ONE. Congratulations! Your manuscript is now being handed over to our production team.

Kind regards, 

on behalf of

Prof. Taye Beyene Demissie 

Academic Editor

PLOS ONE